# Plasma microRNA Signature as Companion Diagnostic for Abiraterone Acetate Treatment in Metastatic Castration-Resistant Prostate Cancer: A Pilot Study

**DOI:** 10.3390/ijms25115573

**Published:** 2024-05-21

**Authors:** Simone Detassis, Francesca Precazzini, Margherita Grasso, Valerio Del Vescovo, Francesca Maines, Orazio Caffo, Paola Campomenosi, Michela A. Denti

**Affiliations:** 1Department of Cellular, Computational and Integrative Biology (CIBIO), University of Trento, Via Sommarive 9, 38123 Trento, TN, Italy; simone.detassis@optoi.com (S.D.);; 2OPTOI Srl, Via Vienna 8, 38100 Trento, TN, Italy; 3Istituto Zooprofilattico Sperimentale Delle Venezie, Sezione di Bolzano, Via Laura Conti 4, 39100 Bolzano, BZ, Italy; 4L.N.Age Srl-Link Neuroscience and Healthcare, Via Mario Savini 15, 00136 Roma, RO, Italy; 5Kapadi Italy Srl, Corso Italia 22, 20122 Milano, MI, Italy; 6Division of Oncology, Santa Chiara Hospital, Largo Medaglie D’oro 9, 38122 Trento, TN, Italy; 7Department of Biotechnology and Life Sciences (DBSV), University of Insubria, Via J.H. Dunant 3, 21100 Varese, VA, Italy

**Keywords:** circulating microRNAs, plasma, prostate cancer, abiraterone acetate, companion diagnostic

## Abstract

Abiraterone acetate (AA) serves as a medication for managing persistent testosterone production in patients with metastatic castration-resistant prostate cancer (mCRPC). However, its efficacy varies among individuals; thus, the identification of biomarkers to predict and follow treatment response is required. In this pilot study, we explored the potential of circulating microRNAs (c-miRNAs) to stratify patients based on their responsiveness to AA. We conducted an analysis of plasma samples obtained from a cohort of 33 mCRPC patients before and after three, six, and nine months of AA treatment. Using miRNA RT-qPCR panels for candidate discovery and TaqMan RT-qPCR for validation, we identified promising miRNA signatures. Our investigation indicated that a signature based on miR-103a-3p and miR-378a-5p effectively discriminates between non-responder and responder patients, while also following the drug’s efficacy over time. Additionally, through in silico analysis, we identified target genes and transcription factors of the two miRNAs, including PTEN and HOXB13, which are known to play roles in AA resistance in mCRPC. In summary, our study highlights two c-miRNAs as potential companion diagnostics of AA in mCRPC patients, offering novel insights for informed decision-making in the treatment of mCRPC.

## 1. Introduction

Prostate cancer (PCa) ranks among the most prevalent cancers in males [1], with risk factors including age, obesity, ethnicity, and family history [2,3]. Genetic alterations in genes such as *BRCA1*, *BRCA2*, *TP53*, androgen and Vitamin D receptors, *HPC1*, *HPC2*, *HPCX*, *CAPB*, *MLH1*, *MSH2* and *MSH6*, *CHEK2*, *PMS2, HOXB13*, *ATM*, and *TMPRSS2-ERG* and *TMPRSS2-ETV1/4* fusions, have been associated to prostate cancer [2,4,5,6,7,8,9]. PCa encompasses a spectrum of malignant tumors, including the most frequent adenocarcinomas, originating in prostate glands and ducts, squamous cell carcinomas, and small-cell prostate cancer, of which a minority is represented by neuroendocrine tumors [10,11,12]. The typical progression of cancer in the prostate starts with prostatic intraepithelial neoplasia [13], where PCa remains confined while the basal layer of the prostate is unharmed. However, high-grade lesions often invade the stroma, leading to metastatic tumor formation [14], with *PTEN* loss and *MYC* amplification recognized as key drivers of high-grade PCa and metastasis [15,16,17,18]. Currently, diagnosis and prognosis of PCa rely on imaging methods and biochemical increase and recurrence of prostate-specific antigen (PSA) [19].

While early stage PCa is often responsive to androgen deprivation [20], advanced stages may develop resistance to this treatment, a phenomenon termed castration-resistant prostate cancer (CRPC). The treatment landscape for metastatic CRPC (mCRPC) has evolved significantly, with several agents approved for clinical use in the US and Europe, demonstrating improved survival rates in both treatment-naïve and previously treated mCRPC patients [21]. Among these strategies, abiraterone acetate (AA), a cytochrome P450 c17 (CYP17A) inhibitor, has shown efficacy in mCRPC patients, including those who have received chemotherapy [21,22]. AA functions by inhibiting testosterone production from testicular, adrenal, and prostate cancer tissues [23]. However, its efficacy varies among patients, and concerns regarding adverse effects such as hypertension, hypokalemia, peripheral edema, and liver enzyme elevation underscore the importance of avoiding unnecessary treatments [24]. Consequently, the need for companion diagnostics to improve clinical decision-making has become increasingly evident.

In the last two decades, microRNAs (miRNAs) have emerged as potential biomarkers for many kinds of pathological conditions, from neurodegenerative diseases [25,26,27] to cancer, and for prognosis and prediction of pharmaceutical treatment efficacy [28,29,30,31,32,33]. MiRNAs, a class of small non-coding RNAs, modulate gene expression post-transcriptionally by promoting mRNA degradation or inhibiting translation [34]. Numerous studies suggest that miRNAs are sensitive to physiological or pathological alterations, rendering them specific to the disease under investigation. They not only have the potential to signal the presence of the disease before clinical symptoms emerge but also to predict treatment outcomes and prognosis [28]. They possess many attributes of an ideal biomarker, meeting both analytical criteria and clinical usefulness. As nucleic acids, miRNAs are relatively straightforward to quantify, with RT-qPCR often serving as the gold-standard method [35]. In this context, circulating miRNAs (c-miRNAs) became a valuable source [36,37] due to (1) the minimally invasive procedure to obtain them from body fluids, (2) great stability, and (3) resistance to ribonucleases and to the harsh physicochemical conditions [38]. These features make c-miRNAs suitable for clinical applications [28]. The present work describes a prospective pilot study on a cohort of mCRPC patients treated with AA, finding indications of a c-miRNA signature able to predict and follow over time the response to AA treatment.

## 2. Results

### 2.1. Patients’ Cohort

The patients’ cohort showed no statistical difference in the median age between responder (R) and non-responder (NR) patients (all = 75.0 ± 7.5; NR = 75.0 ± 8.55; 74.5 ± 5.0). Prior to AA administration, 21 out of 21 NR patients and 11 out of 12 R patients received docetaxel, 11 out of 22 NR patients and 2 out of 11 R patients received enzalutamide, and 8 out of 21 NR patients and 6 out of 12 R patients received cabazitaxel. The median PFS (months) of NR patients was 3.9 ± 1.0 and 14 ± 4.9 for R patients (Appendix A).

### 2.2. Candidate miRNAs Discovery through miRNome Analysis

miRNome analysis via human Exiqon qPCR panels—evaluating 752 miRNAs—was used for the discovery phase. Plasma was sampled from 9 R and 9 NR patients prior to AA administration (preTRT) and analyzed (Figure 1). 319 miRNAs were excluded from the analysis due to non-detectable levels in more than 20% of the samples; 198 miRNAs were present in all the samples; 2 miRNAs were non-detectable in the whole NR group. Hemolysis was assessed according to criteria described by Blondal et al. [39] (Appendix A). Expression data were normalized on miR-425-5p, as determined by Normfinder [40] (Appendix A). Based on the expression analysis of miRnome panels and a literature review [41,42,43,44,45,46,47,48,49,50,51,52,53,54,55,56,57,58,59,60,61,62,63,64], 7 miRNAs (miR-144-3p; miR-182-5p; miR-29b-3p; miR-331-5p; miR-33a-5p; miR-363-3p; miR-378a-5p) were selected as candidates for the validation phase (Appendix A). To this panel, we added miR-103a-3p, selected from the literature review [41,42,43,44,45,46,47,48,49,50,51,52,53,54,55,56,57,58,59,60,61,62,63,64]. It is important to highlight that miR-103a-3p did not show differential expression between NR and R patients in our initial miRNome analysis (Figure 1). However, in the literature review conducted, the role of miR-103a-3p in prostate cancer progression during androgen ablation therapy and chemotherapy [41,42,43,44] was significant enough to include it in validation with a larger sample set. Specifically, its validated regulatory axis with PTEN (miRTarBase V9.0 data [65]) was a key consideration, as the loss of PTEN is linked to resistance to androgen ablation therapy [66,67].

### 2.3. Validation Analysis on Eight miRNA Candidates Highlights Two miRNAs Predicting AA Efficacy in mCRPC

The eight miRNA candidates were validated using TaqMan RT-qPCR on the same preTRT samples utilized in the discovery phase (Figure 2). miR-33a-5p was excluded from further analyses because of its undetectable expression levels in four R patients and Cq levels above 37 in the others. This made miR-33a-5p unsuitable as a robust biomarker and difficult to conduct proper statistical analysis on its expression. During RT-qPCR validation, miR-103a-3p also showed a significant difference in levels in the two groups of patients. Among the candidates, miR-144-3p, miR-182-5p, miR-331-5p, miR-363-3p, and miR-378a-5p exhibited expression patterns consistent with the miRNome analysis. However, only miR-103a-3p and miR-144-3p showed a significant difference between the two patient groups.

To improve discrimination capacity, we evaluated combined signatures of two miRNAs to identify a single combination that effectively enhanced discrimination between NR and R groups. A miRNA score (miRS) comprising the ratio of miR-103a-3p and miR-378a-5p was the most effective in distinguishing NR from R patients, as illustrated in Figure 3A (*p*-value = 0.0001). Notably, the miRS, being a ratio of the two miRNA expression levels (see Secction 4.6 in Materials and Methods), does not need the use of a normalizer such as miR-425-5p. To validate these findings, the dataset was expanded to include 15 additional patients (12 NR and 3 R), resulting in a total of 21 NR and 12 R patients. As depicted in Figure 3B,C, miRS confirmed its ability to distinguish the two patients’ groups (AUC = 0.81; *p*-value: 0.008). As the cohort previously underwent other treatments involving enzalutamide and cabazitaxel (docetaxel was not included, as all patients but one had received this treatment), additional analyses were carried out to exclude potential interferences of the drugs with miRS (Appendix A). These results suggest that miRS holds promise as a potential predictive biomarker of AA efficacy in mCRPC patients.

### 2.4. Analysis of Follow-Up Samples Suggests miRS as a Companion Diagnostic for AA

The efficacy of AA over time was assessed using miRS. Post-treatment plasma samples were obtained from the same NR and R patients, with varying availability across different time points (11 R and 14 NR at 3 months; 10 R and 4 NR at 6 months; 7 R and 0 NR at 9 months). The unavailability of certain samples was due to patient death or termination of treatment. As illustrated in Figure 4, miRS appears to provide information on the responsiveness to AA over time. It is worth noting that the miRS of R patients progressively resembles that of NR patients, potentially serving as an indicator of the real-life scenario of a shift toward resistance to AA.

To mitigate potential confounding factors coming from post-treatment clinical classification, an unsupervised clustering model was employed using miRS data to group patients. This approach successfully classified patients into two distinct groups shown in Figure 5A (groups A and B; *p*-value < 0.0001). Figure 5B depicts the Kaplan–Meyer plot, which reveals a significant difference (*p*-value = 0.0003) between group A (median PFS = 9 months) and group B (median PFS = 3.8 months). Accordingly, group A was interpreted as responder patients, while group B as non-responder patients. Notably, compared to the clinical classification, the unsupervised model reclassified five clinically classified NR patients as group A (responders), and two clinically classified R patients as group B (non-responders). The clustering of post-treatment samples by the model exhibited a consistent trend as observed in previous analyses (Figure 5C). Furthermore, to demonstrate miRS ability to monitor AA responsiveness over time, the difference between miRS at preTRT and miRS at 3 months in R patients was assessed (termed “delta score”; 2 out of 15 samples lacked 3-month analysis). Figure 5D suggests a negative correlation between the delta score and PFS. It can be speculated that miRS may identify changes in AA responsiveness after 3 months of treatment in patients classified as responders. Notably, the five reclassified patients in the responder group (clinical classification was NR) exhibited PFS values of 5.5, 5.5, 4.7, 4.6, and 3.8 months, all higher than or equal to the median PFS of the non-responder group (B group; 3.8 months). While miRS classified them as responders, they may be subgrouped as “poor responders”. Intriguingly, three of these patients (highlighted in Figure 5D) also displayed a high delta score, inversely correlated with PFS. This suggests a hypothetical clinical scenario where miRS after 3 months might have signaled the necessity to discontinue AA treatment for these individuals.

### 2.5. In Silico Analysis of miR-103a-3p and miR-378a-5p Targets and Promoters

To understand the possible roles of the chosen miRNAs, an analysis of predicted and validated targets was conducted. Notably, several genes associated to prostate cancer advancement, recurrence, and therapy resistance emerged as the most relevant targets. These included WNT2B and WNT7A (predicted targets for miR-103a-3p) [68,69,70], DICER1 [71], PTEN (validated target for miR-103a-3p) [67], and SP1 (validated target for miR-378a-5p) [72,73,74]. Gene ontology analysis on validated targets of both miRNAs (Appendix A) revealed enrichment in cell cycle pathways and p53 signaling. Additionally, GeneXplain software was used to investigate possible consensus sequences of human transcription factors enriched in the promoters of miR-378a-5p and miR-103a-3p. Four different Ensembl annotations were examined: miR-378a-5p, miR-103a1-3p, miR-103a2-3p, and PANK2. miR-103a is present as miR-103a1 and miR-103a2 in the human genome, expressing two identical mature miRNAs. Neither TaqMan RT-qPCR probes nor the target predictions algorithms can discriminate between the two identical sequences. However, being present in two different genomic locations, transcription factor analysis might lead to different results. Additionally, miR-103a2-3p is located within one of the introns of the *PANK2* gene. Thus, the *PANK2* promoter was also selected for this analysis. The transcription factors with enriched consensus binding sequences for both miR-103a-3p and miR-378a-5p were ATF2, BRCA1, GMEB2, IRX2, LEF1, MAFA, SOX10, USF2, ZBTB33, and HOXB13. Notably, HOXB13, particularly enriched in miR-103a-3p promoters, has previously been associated with early mortality following AA treatment [75].

## 3. Discussion

The evolving therapeutic landscape of metastatic castration-resistant prostate cancer (mCRPC) over the last decade has represented a challenge for clinicians, who must navigate through various treatment options [21]. However, given that all patients eventually develop primary or acquired resistance to available drugs, the urgent need for effective biomarkers to identify individuals who are most likely to benefit from specific treatments is crucial. While a few studies have explored biomarkers for the efficacy of AA, such as androgen receptor variant AR-V7, the TMPRSS2-ERG fusion gene, and PTEN [76,77,78,79], robust evidence remains limited. Currently, only the FoundationOne^®^CDx diagnostic test is FDA-approved for guiding AA treatment, specifically in BRCA1–2-mutated mCRPC patients [80]. Hence, there is a critical need to discover new companion diagnostics for AA to align with clinical demands.

microRNAs (miRNAs) have emerged as promising biomarker candidates due to their potential for diagnosis, prognosis, and predicting drug efficacy. Engaged in post-transcriptional regulation, miRNAs play roles in various biological processes and exhibit sensitivity to physiological and pathological changes [28,30]. Despite technological challenges, measuring miRNAs via RNA sequencing or RT-qPCR is relatively straightforward [35]. Particularly, circulating miRNAs (c-miRNAs) found in biofluids like plasma, serum, or urine offer significant potential. Although susceptible to confounding factors such as hemolysis or sampling variability [28,81], liquid biopsy procedures are less invasive than those of solid biopsies, enhancing the clinical utility of circulating biomarkers. Furthermore, c-miRNAs are stable in biofluids, showing resistance to nucleases and environmental stresses [38]. While the expression of c-miRNAs as predictive biomarkers in mCRPC patients has been investigated [82,83,84,85], limited research has specifically targeted the prediction of AA efficacy [86,87], and, to date, none have successfully predicted and monitored AA efficacy over time. 

In this pilot study, plasma samples were collected from 33 mCRPC patients treated with AA, before treatment initiation and at 3, 6, and 9 months post-administration. A comprehensive profiling of c-miRNAs using RT-qPCR analysis was conducted. Following candidate analysis utilizing the miRNome Exiqon panel and a literature review, eight miRNAs were selected for further investigation. During the validation phase, a scoring system incorporating two miRNAs, miR-103-3p and miR-378a-5p (miRS), emerged as capable of predicting the response to AA treatment and detecting disease progression during therapy. Employing an unsupervised clustering model to mitigate confounding factors from clinician classification, responder and non-responder patients were effectively grouped based on miRS data. The model retrospectively reclassified five patients clinically evaluated as non-responders into the responders’ group, and notably, they exhibited a PFS higher than the average of the non-responder group. These findings suggest that miRS may offer greater sensitivity to AA efficacy compared to clinicians’ classification, which relies on parameters like PSA progression and clinical or radiological indicators of disease progression. Moreover, within the responders’ group, the difference between miRS levels in pre-treatment samples and at 3 months (delta score) suggests a negative correlation with PFS, indicating that miRS might track AA efficacy over time. This has significant implications for clinicians, aiding in determining whether to continue or suspend the treatment. Furthermore, three of the reclassified patients in the responders’ group displayed a high delta score and low PFS, indicating that miRS not only classified them as responders (with higher PFS compared to classified NR) but also signaled the need to discontinue treatment after 3 months.

Bioinformatic analysis of targets and promoters provides evidence supporting the involvement of the two miRNAs in primary or acquired resistance to AA treatment. Notably, PTEN is a validated target of miR-103-3p, and studies have demonstrated that PTEN loss negatively impacts prognosis after AA treatment [67]. It is plausible that altered levels of miR-103a-3p could influence PTEN expression, thereby increasing cancer aggressiveness. Similarly, SP1, a validated target of miR-378a-5p, has been shown to positively regulate the expression of CYP17A1, the target of AA [88]. It can be hypothesized that alterations in miR-378a-5p levels may disrupt the SP1/CYPA17A1 axis, thereby affecting AA efficacy. Promoters’ analysis highlighted enriched transcription factor binding sites for developmental genes, such as HOX and SOX family genes (HOXB13 and SOX10), and for well-known cancer-related genes, like BRCA1. Interestingly, HOXB13 is highly expressed in circulating tumor cells and predicts early death following AA treatment [75]. Similarly, BRCA genes have been associated with a high risk of prostate cancer and poor prognosis [89,90]. While this pilot study involves a small patient cohort, miRS data indicate a potential new companion diagnostic for AA, currently absent in clinical settings. Further investigation with a larger sample size is necessary, along with an examination of the potential molecular mechanisms of action of miR-103a-3p and miR-378a-5p in the responsiveness of mCRPC patients to AA.

## 4. Materials and Methods

### 4.1. Patients’ Cohort

A cohort of 33 mCRPC patients treated with abiraterone acetate (AA) as per standard clinical practice at Santa Chiara Hospital (Trento, Italy) from 2013 to 2015 was recruited. Written informed consent was obtained from all subjects participating in the study. The protocol followed the principles of the Declaration of Helsinki, and it was approved by the Ethical Committee of the Trentino Local Health Authority (Comitato Etico per le Sperimentazioni Cliniche, Azienda Provinciale per i Servizi Sanitari, Protocol ID #42362666). The treatment consisted of daily oral administration of 1000 mg AA plus 5 mg prednisone [91]. During the treatment period, patients underwent a monthly clinical assessment with regular evaluation of bone marrow, renal, and liver functions and measurement of PSA expression. Radiological evaluation of the tumor was executed every three months or whenever clinically suggested. The treatment was maintained until disease progression, which followed PCWG2 (Prostate Cancer Working Group 2) guidelines [92] as the evidence of clinical (performance status deterioration) or radiological progression. AA administration was not interrupted only at the presence of PSA increase. The patients were classified AA responders in the evidence of (1) radiological change without PSA progression and performance status deterioration or (2) in the presence of PSA reduction ≥ 50% compared to the pre-treatment values without radiological progression and performance status deterioration. Twelve patients showed responsiveness to AA, while the remaining 21 patients were classified as non-responders to AA. Appendix A shows patients’ details.

### 4.2. Plasma Samples Processing

Upon informed consent signature, a 5 mL blood sample was obtained from the patients before starting administration of AA and every three months until treatment termination as per standard guidelines at Santa Chiara Hospital. Immediately after the collection in EDTA tubes, the blood samples were centrifuged at 2500 rpm (4 °C) for 15 min. The plasma was carefully collected without disturbing the buffy coat and stored in 250–500 µL aliquot parts at the Trentino Biobank at −80 °C (Santa Chiara Hospital, Trento, Italy). 

### 4.3. RNA Extraction and Quantification

Then, 250 µL of the plasma samples were processed for the extraction of total RNA using the miRNeasy^®^ Serum/Plasma Kit (Qiagen, Hilden, Germany). Briefly, samples were lysed by adding 5 volumes of QIAzol Lysis Reagent containing 1 µg of MS2 phage RNA. Following manufacturer’s protocol, chloroform was added for extrapolating the aqueous phase, which was loaded onto RNeasy Mini spin columns. All the subsequent washing steps were performed as per the manufacturer’s protocol. Total RNA was eluted in 40 µL of nucleases-free water. Accurate quantification of the 18 samples employed for Exiqon miRNome RT-qPCR panels (Qiagen) was performed by the Qubit fluorometer with a Qubit RNA HS Assay Kit (Thermo Fisher, Waltham, MA, USA). 

### 4.4. Exiqon miRNome RT-qPCR Panels

Retrotranscription by Universal cDNA synthesis kit (Exiqon—Qiagen) was performed starting from 80 ng of extracted RNA mixed with 4 µL of 5X reaction buffer, 2 µL of enzyme, and nucleases-free water (total volume = 20 µL). The assay was executed in a thermocycler (T100 Thermal Cycler, BioRad, Hercules, CA, USA) as per the kit’s instructions: 60 min at 42 °C and 5 min at 95 °C. miRNAs screening was performed with an Exiqon microRNA Ready-to-Use PCR (miRCURY LNA Universal RT microRNA PCR human panel I + II, V4.M—Qiagen) employing cDNA template diluted 1:50 in nucleases-free water and 2X SYBR Green Universal PCR Master Mix (Exiqon—Qiagen) mixed 1:1 (total volume = 10 µL). The reaction was executed in a thermocycler (CFX384 Real-Time PCR Detection System, BioRad) as per the kit’s instructions: 10 min at 95 °C and 40 amplification cycles of 10 s at 95 °C and 1 min at 60 °C.

### 4.5. TaqMan RT-qPCR

TaqMan RTq-PCR was employed for single miRNA validations. Retrotranscription (TaqMan^®^ microRNA Reverse Transcription Kit, Thermo Fisher) was performed with 2 µL of extracted RNA mixed with 0.19 µL of RNase inhibitor, 1.5 µL of 10X reaction buffer, 0.15 µL of dNTPs mix, 1 µL of enzyme, and 3 µL of TaqMan^®^ microRNA assay miR-103a-3p (#000439), miR-182-5p (#002334), miR-144-3p (#197375), miR-29b-3p (#000413), miR-33a-5p (#002135), miR-331-5p (#002233), miR-363-3p (#001271), miR-378a-5p (#000567), miR-425-5p (#001516), and nucleases-free water (total volume = 15 µL). The qPCR step was performed with 1.33 µL of cDNA product mixed with 5 µL of Faststart TaqMan^®^ Buffer (Sigma Aldrich), 1 µL of TaqMan^®^ microRNA assay (listed above), and nucleases water (total volume = 10 µL). Retrotranscription and qPCR were executed in a thermocycler (Biorad CFX 384) as per the kit’s instructions: 30 min at 16 °C, 30 min at 42 °C, 5 min 85 °C for retrotranscription, and 10 min at 95 °C, 40 amplification cycles of 15 s at 95 °C and 1 min at 60 °C for qPCR.

### 4.6. Data Analyses

Exiqon microRNA Ready-to-Use PCR panels data were processed and analyzed with GenEX software (version 6). miRNAs non-detected in at least 20% of the samples were removed, and Ct equal or higher than 40 were considered non-detected. miR-425-5p was selected by NormFinder software (version 0.953) [40] as a normalizer of Exiqon panels data. The volcano plot fold change was generated evaluating the NR group relative to the R group. For validation analysis, TaqMan RT-qPCR Ct higher than or equal to 40 were considered non-detected. TaqMan RT-qPCR normalization was performed on miR-425-5p. The miRNA score (miRS) was calculated as follows: 2^−[Cq(miR-103-3p) − Cq(miR-425-5p)]^/2^−[Cq(miR-378a-5p) − Cq(miR-425-5p)]^ = 2^−[Cq(miR-103-3p) − Cq(miR-425-5p)] − {−[Cq(miR-378a-5p) − Cq(miR-425-5p)]}^= 2^−Cq(miR-103-3p) + Cq(miR-425-5p) + Cq(miR-378a-5p) − Cq(miR-425-5p)^ = 2^Cq(miR-378a-5p) − Cq(miR-103-3p)^. For statistical analyses on Exiqon and TaqMan RT-qPCR, Student’s *t*-test was employed (with Bonferroni correction for Exiqon panels’ data). The Kaplan–Meyer plot and the receiver operating characteristic curve were analyzed by GraphPad Prism V8. Unsupervised clustering was generated via R software (V 3.4.1) and RStudio (V 1.0.143) with the “rpart” package (V 4.1-11).

### 4.7. In Silico Analyses

The predicted targets of miR-103a-3p and miR-378a-5p were recovered, merging TargetScan (version 7.2) [93], Pita (version 6) [94], and MiRanda software (version 0.95.6.18429 Final R55) [95] filtering for binding energy ≤ −18 kcal/mol. Validated targets were retrieved from the miRTarBase database (V9.0) [65]. Gene ontology analysis on validated targets was performed by Metascape (version 3.5.20240101) [96]. The promoters of miR-378a-5p, miR-103a1-3p, miR-103a2-3p, and PANK2 (ENSG00000199047 (miR-378a), ENSG00000199035 (miR-103a1), ENSG00000199024 (miR-103a2), and ENSG00000125779 (PANK2)) were investigated for the transcription factor analysis searching from −4900/−3900/−2900/−1900/−900 to +100 (transcription start site at +1, retrieved automatically by GeneXplain software TRANSFAC 2.0 from miRBase V21).

## Figures and Tables

**Figure 1 ijms-25-05573-f001:**
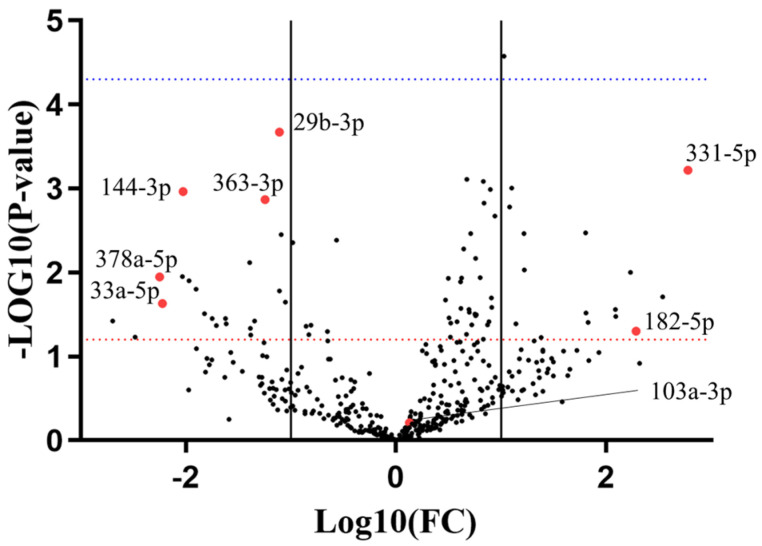
Exiqon miRNome panels analysis with a volcano plot. X axis refers to the non-responder (NR) versus responder (R) group. Solid lines represent 1 < log(Fold Change) < −1. Red dotted line represents *p*-value = 0.05, blue dotted line represents *p*-value = 0.0001. Red dots represent miRNA candidates chosen for the validation phase.

**Figure 2 ijms-25-05573-f002:**
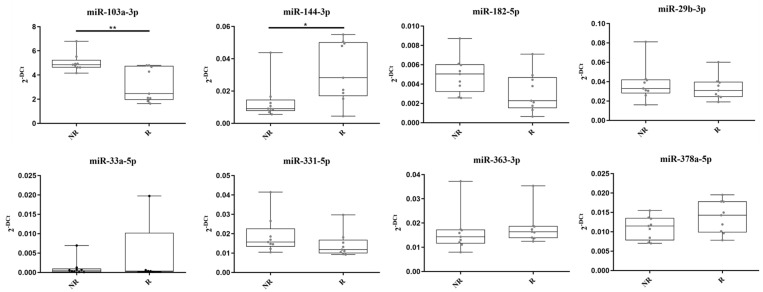
TaqMan RT-qPCR validation of eight miRNA candidates. * = *p*-value < 0.05. ** = *p*-value < 0.01. miR-33a-5p expression levels were non-detectable in four samples of the responder group.

**Figure 3 ijms-25-05573-f003:**
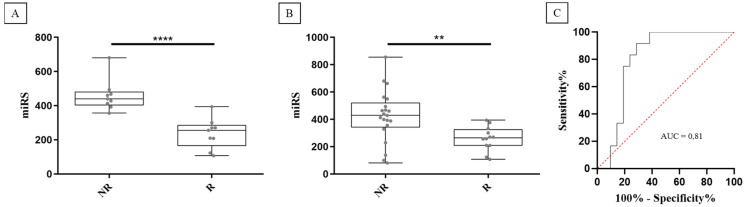
miRS significantly discriminates between non-responder (NR) and responder (R) patients in both the original cohort (**A**) and in the expanded cohort (**B**). ROC curve analysis shows a good accuracy for predicting responder patients (red dotted line represents the random guess) (**C**). **** = *p*-value < 0.001. ** = *p*-value < 0.01.

**Figure 4 ijms-25-05573-f004:**
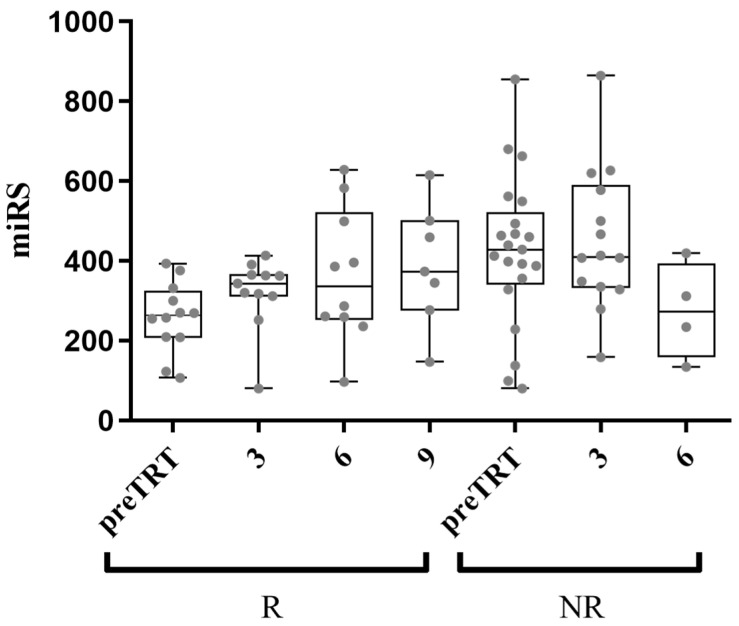
Follow-up samples’ analysis with miRS at 3, 6, and 9 months after AA administration. The average miRS of responder (R) patients shows a trend progressively comparable to the average non-responder (NR) patients’ miRS, possibly indicating the gradual acquisition of resistance to AA (*x*-axis shows preTRT and months after treatment).

**Figure 5 ijms-25-05573-f005:**
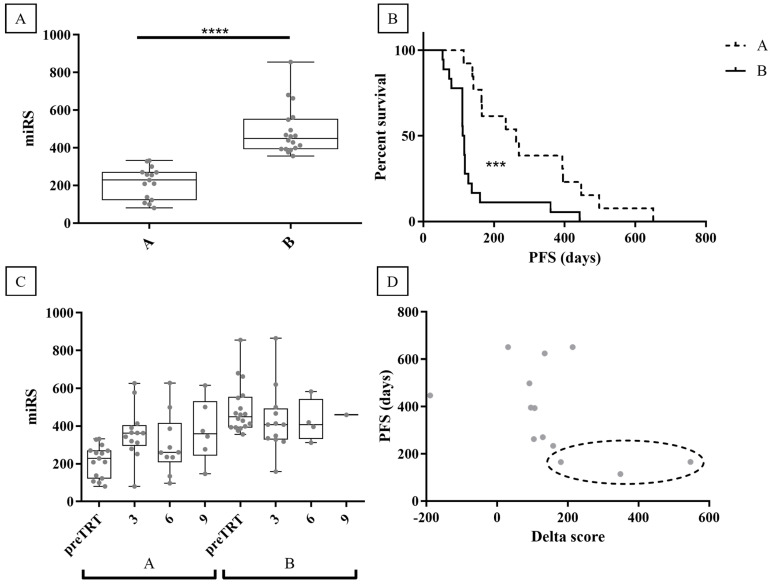
Unsupervised clustering of miRS data. (**A**) K-means clustering of miRS divides the patients into two groups (A -B). (**B**) Kaplan -Meyer plot of the clustered data defines group A as responders and group B as non-responders. (**C**) Follow-up samples’ analysis with miRS at 3, 6, and 9 months after AA administration on the clustered groups. (**D**) The delta score of group A (responders) suggests a negative correlation with PFS. Reclassified patients are highlighted in the dotted line oval. **** = *p*-value < 0.0001, *** = *p*-value < 0.001.

## Data Availability

Exiqon miRNome panels have been submitted to GEO (GSE262550). Other data are available upon request.

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
