# Peer review of "Plasma microRNA Signature as Companion Diagnostic for Abiraterone Acetate Treatment in Metastatic Castration-Resistant Prostate Cancer: A Pilot Study"

_ijms, 2024, doi:10.3390/ijms25115573_

Round 1

Reviewer 1 Report

Comments and Suggestions for Authors

April 8, 2024

Ms. Ref. No.: ijms-2962670

Journal: International Journal of Molecular Sciences.

Title: Plasma microRNA Signature as Companion Diagnostic for Abiraterone Acetate Treatment in Metastatic Castration-Resistant Prostate Cancer: A Pilot Study.

Comments:

Thank you for your efforts in composing an article on such a pertinent subject. I have taken the liberty of providing you with a few observations that I believe will serve to enhance the quality of your work. Please find my feedback outlined in the following paragraphs

1-      In this study, there are three durations times treatment with Abiraterone Acetate that are three, six, and nine months, how is this interval?

2-      The sample size of this study is 33, the next question is about how to calculate this sample size.

3-      The treatment in this study involves AA (1000 mg) plus prednisone 5 mg twice a day, why prednisone? And why 5 mg for it and 1000 mg for AA?

4-      The main factor for detecting the effect of this treatment was PSA or radiological progression. and why?

5-      The next question is about results. The research was conducted in two different steps, the first one was molecular tests such as PCR and …, and the second was in-silico tests. While the results for the first tests were about miR-103a-3p and miR-378a-5p the second tests were about PTEN and HOXB13, are these differences analyzed factors reasonable?

6-      In some figures such as Fig 2, 3, 4, and 5, are the R-Squared Ok?

7- To improve the clarity of the introduction, it is recommended that you include some of the following sources as references:

·         https://doi.org/10.3390/ijms14035264

·         https://doi.org/10.1007/s00210-023-02551-0

Author Response

Dear reviewer,

we thank you for taking the time and effort to review our manuscript and to provide valuable comments and suggestions. In the attached file, we reply one by one to your comments and questions.

Reviewer 2 Report

Comments and Suggestions for Authors

Destasis et al. aimed to characterize a plasma microRNA signature of Abiraterone Acetate (AA) responsiveness in prostate cancer patients after docetaxel failure. They have designed a two-step approach, with a discovery screening step based on Exiqon microarrays and a validation step based on TaqMan assays, followed by a bioinformatic analysis of the identified microRNAs' targets. 

Major concerns

1. The authors claimed to have used a "homogeneous series of 33 mCRPC patients who were treated with abiraterone acetate (AA) after docetaxel failure". However, Table S1 shows that, besides docetaxel, the patients enrolled were treated with enzalutamide (6 of the responders, none of the responders) and cabazitaxel (8 of the responders, 6 of the nonresponders). Of note, the data in Table S1 differ from those in the main text (lines 84/85). Furthermore, Table S1 shows that the patients were exposed to different drug combinations (DCX, DCX+ENZ, DCX+CBX, DCX+ENX+CBX) before the onset of AA therapy; hence, the two groups of patients (responders and nonresponders) were already significantly different in terms of their previous exposure to these three drugs. Given the well-documented impact of all these molecules on microRNA expression, I am afraid that the authors' results cannot be attributed solely to the AA clinical-responder status.

2. Although not picked up by their discovery approach, the authors have decided to include in their validation analysis miR-103a-3p simply because of "its documented role in prostate cancer progression during androgen ablation therapy and chemotherapy". Why they chose this particular microRNA is unclear, as it is unclear why the authors haven't screened the literature for other microRNAs to complement their discovery step.

3. The Exiqon screening approach was based on an amplification protocol with 40 cycles (line 340), the usual norm. However, in the validation step, the threshold considered was 37 (line 115), which led to miR-33a-5p invalidation. I fail to understand the differences between the Cq-threshold values used in the two steps.

4. The authors state that only two microRNAs, miR-144-3p and miR-103a-3p, showed a significant difference between responders and nonresponders (line 119). However, their analysis continues with a combined score (named miRS), defined as the ratio between miR-103a-3p (statistically differentially expressed) and miR-378a-5p (statistically not differentially expressed). I am afraid I fail to understand the logic behind this approach and why miR-144-3p was "left behind".

5. I find the authors' decision to use unnormalized data on miR-103a-3p and miR-378a-5p for miRS calculation extremely troubling ("Notably, the miRS does not need the use of a normalizer such as miR-425-5p"). I am afraid that the use of unnormalized Ct values (in any relative quantification of microRNA expression) is not acceptable, and hence, all the subsequent analyses (correlations on follow-up samples, Kaplan-Meyer, the unsupervised model that contradicted the clinical stratification in responders and nonresponders) based on these miRS are faulty (and their clinical interpretations highly speculative).

Comments on the Quality of English Language

Minor editing of the English language is required.

Author Response

(The authors gave the same response as above.)

Reviewer 3 Report

Comments and Suggestions for Authors

Title - precisely summarizing the essence of the study - No remarks

Abstract - concisely summarizing the study - No Remarks

Introduction - in-depth presentation of the background on the obstacles in personalizing treatment in mCRPC, emphasizing on c-miRNA as biomarkers - No remarks

Material and methods - comprehensive description of the highly sophisticated study protocol - No Remarks

Results - step-wise results form preliminary candidate miRNA discovery, validation of 8 potential miRNA, finding miR-144-3p and miR-103a-3p and their ratio (miRNA score (miRS)) as a strong discriminator between responders and non-responders. The authors took it even further researching on those miRNA in the course of the treatment as leads to developing AA resistance. They took another step in connecting miRNA to genes, searching for their targets and promoters - No remarks

Discussion - strongly substantiated conclusions on the potential use of miRNA score (miRS) as a tool for stratification and follow-up regarding abiraterone treatment in mCRPC - No Remarks

Author Response

Dear reviewer,

we thank you for taking the time and effort to review the manuscript. We appreciate that it was clear and properly written.

Round 2

Reviewer 1 Report

Comments and Suggestions for Authors

no thanks